# Real-time Mapping of Physical Scene Properties with an Autonomous Robot Experimenter

**Iain Haughton**[1]  **Edgar Sucar**[2]  **Andre Mouton**[1]  **Edward Johns**[3]  **Andrew J. Davison**[2]

[1] Dyson Technology Ltd.
[2] Dyson Robotics Lab, Imperial College
[3] Robot Learning Lab, Imperial College
`iain.haughton@dyson.com`

**Abstract:**
Neural fields can be trained from scratch to represent the shape and appearance of 3D scenes efficiently. It has also been shown that they can densely map correlated properties such as semantics, via sparse interactions from a human labeller. In this work, we show that a robot can densely annotate a scene with arbitrary discrete or continuous physical properties via its own fully-autonomous experimental interactions, as it simultaneously scans and maps it with an RGB-D camera. A variety of scene interactions are possible, including poking with force sensing to determine rigidity, measuring local material type with single-pixel spectroscopy or predicting force distributions by pushing. Sparse experimental interactions are guided by entropy to enable high efficiency, with tabletop scene properties densely mapped from scratch in a few minutes from a few tens of interactions.

**Keywords:** neural field, robot experimentation, physical properties

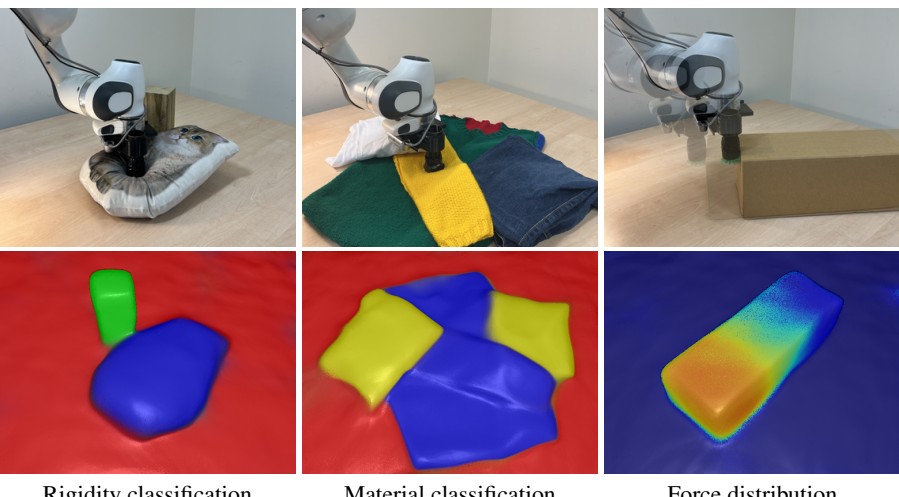

Rigidity classification    Material classification    Force distribution

Figure 1: Over a few minutes our robot makes sparse, automatic physical scene interactions, such as touching to test rigidity, sampling local material type with spectroscopy or pushing to determine frictional force distribution. The interaction results are used as sparse labels to the output channels of a joint neural-field model of 3D shape and appearance, trained in real-time. Model coherence allows the measured physical properties to be efficiently and densely propagated to the whole scene, without the need for prior training data.

## 1 Introduction

An autonomous agent performing complex tasks in unpredictable environments (e.g. tidying a home), must build a rich internal representation of its environment. This representation should extend

6th Conference on Robot Learning (CoRL 2022), Auckland, New Zealand.

beyond geometry, colour and traditional semantics to include physical properties from which robotic affordances may be inferred. Like its biological counterparts, a robotic agent cannot infer these properties from vision alone [1]. In this work, we present a real-world robot that learns about its environment by autonomous exploration and experimentation. The robot incrementally discovers physical scene properties by performing targeted physical interactions with the scene, measuring their effects and feeding the results into a jointly-optimised neural-field MLP. With only a handful of these autonomous robotic experiments, a task-driven internal representation of the scene is built from scratch, without any pre-training or external guidance.

We build upon the observation that the joint internal representation of shape and appearance learned by implicit scene representation models, and the smoothness and compactness priors present in these models, allow for ultra-efficient, scene-wide propagation of user-provided labels [2]. We remove the human from the loop entirely, presenting the first fully-autonomous, 3D scene understanding robot that combines active exploration and physical experimentation with a unified neural-field representation as its underlying computational model.

The robot explores a scene with an RGB-D camera and simultaneously builds a map of semantic entropy, representing confidence in its current predictions. Considering this entropy, as well as kinematic feasibility and collision avoidance, the robot selects the optimal point of interaction and proceeds to take a physical measurement used for optimising the semantic head of the underlying MLP. The real-time rendering of entropy guides the robot in selecting efficient and beneficial interactions, allowing for dense scene labelling with only a handful of experiments. In contrast to iLabel [2], which assumes a static scene, and can therefore rely on a growing collection of keyframes over which to optimise semantics, the interactive nature of our framework results in a dynamic scene. In addressing this, we make the observation that neural representations possess temporal memory, and thus propose a new strategy that uses only the latest keyframe during optimisation. In this way, the neural representation remains current, allowing for continuous exploration in scenes with limited dynamics.

Autonomous discovery of physical scene properties potentially facilitates a broad range of practical downstream robotic tasks. For example, automated tool exchange in a surface-cleaning robot; object and material sorting for automated sorting of recycling or laundry; improved task planning enabled by understanding object affordances and physical properties of objects; and improving the safety of autonomous robots by mitigating interactions with potentially hazardous objects or materials.

In summary, the key contributions of this paper are: 1) the first fully-autonomous, neural-scene labelling robot, trained from scratch, in real-time, and capable of operating in the real-world; 2) active robotic experimentation achieved using a single MLP to drive entropy-guided automatic query generation, kinematic feasibility checking, collision avoidance and physical interaction planning and execution; 3) demonstration of the temporal memory characteristics of neural field representations via single-frame optimisation and robustness to small temporal scene changes and 4) prediction of dense, continuous-valued semantics using a neural-field model optimised with respect to sparse ground truths. We substantiate these contributions with a series of real-world experiments, demonstrating the capability of our system to produce a variety of perceptual maps or scene labellings, based on the chosen sensor and action primitives.

## 2 Related work

We address the problem of fully-autonomous, unsupervised scene understanding via active experimentation. Our solution operates at the intersection of several research domains: instance segmentation, interactive perception, affordance learning, and online scene understanding.

**Instance segmentation:** Unseen Object Instance Segmentation (UOIS) is a necessary capability for robots operating in unstructured environments. Xie *et al.* [3] propose a class-agnostic RGB-D instance segmentation method, using only simulated training data with reasonable sim-to-real generalisation. Xiang *et al.* [4] improve sim-to-real transfer by learning RGB-D feature embeddings from synthetic data using a contrastive loss, which allows their Unseen Clustering Network (UCN) to capitalise on both synthetic depth and non-photorealistic RGB data. Gouda *et al.* [5] fine-tune a Mask R-CNN [6] model to perform class-agnostic instance segmentation. The current state-of-the-art in UOIS is achieved by refining the outputs of the UCN method with the graph-based RICE method [7].

**Interactive object segmentation:** Bohg *et al.* [8] first provided a detailed definition of Interactive Perception (IP) and an analysis of its benefits in solving robotic perception tasks. A common approach in interactive object segmentation is to exploit motion cues induced by physical interactions [9, 10, 11, 12]. While our work shares the broad objective of segmentation through interaction, it is not limited to rigid objects or segmenting based on geometric extent. Instead, we demonstrate segmentations that are optimised for given downstream tasks, dictated by the chosen action primitive and sensor.

**Interactive affordance learning:** Le Goff *et al.* [13] learn affordance-specific relevance maps, describing which actions can be applied at which locations. The method relies on an off-the-shelf point-cloud segmentation [14] and an online classifier [15]. Nagarajan and Graumann [16] simultaneously train an interaction exploration policy and an image-based affordance segmentation model mapping image regions to the likelihood that they permit an action. Both these prior works limit experimentation to simulated environments. Although we do not consider affordances explicitly in this work, it is the ultimate objective of performing task-driven segmentation in the way that we do.

**Online scene understanding and labelling:** The iLabel framework [2] (built upon iMAP [17]), demonstrated that online, scene-specific training of a compact MLP model, which jointly encodes scene geometry, appearance and semantics allows for ultra-sparse interactive labelling and produces accurate dense semantic segmentations — outperforming existing pre-trained methods. We remove the human-in-the-loop entirely, demonstrating the first fully-autonomous neural-scene labelling robot. To achieve this, we introduce several significant contributions to allow for operation on a real-world robot, where scene dynamics, kinematics, constraints and interaction feasibility must be considered.

## 3 Method

We represent 3D scenes similarly to iMAP [17], with an MLP that maps a 3D coordinate to colour and volume density. We build a sparse set of keyframes, selected incrementally based on information gain, and whose viewpoints span the scene. The MLP parameters and camera poses are jointly and continually optimised via differential volume rendering of actively-sampled sparse pixels in this set of keyframes. iLabel [2] adds a semantic head to the neural scene MLP in iMAP, allowing users to provide pixel-level annotations in the keyframes. The joint optimisation in iMAP is then extended to include scene semantics, which are optimised through semantic rendering of the user annotations. Both models are trained from scratch, in real-time and without any prior data and have demonstrated the power of the joint internal representations learnt by the neural-field MLP.

### 3.1 Autonomous robot experimentation

The robot builds an internal representation of its environment via a series of autonomous experiments. First, it actively selects interaction locations that are both feasible and information-rich (based on semantic entropy). Second, the selected 2D image locations are mapped to the real-world coordinate system of the robot, and a physical interaction with the scene is planned and executed. Third, the resulting measurement is processed and/or classified (see specifics in Section 3.5) to obtain the ground-truth semantic label (akin to the user-provided object class in [2]). Finally, using the labels obtained in this manner, scene semantics are optimised through semantic rendering of the robot-selected keyframe pixels. As demonstrated previously [2], the resulting joint internal representation of shape, appearance and semantics of the neural-field allows for the sparsely-annotated semantics to be propagated efficiently and densely throughout the scene.

### 3.2 Modes of interaction

Our framework facilitates the autonomous discovery and mapping of any measurable characteristic of a scene, provided that a suitable measurement sensor and interaction protocol can be defined. We demonstrate three particular interaction types: 1) predicting rigidity by top-down poking; 2) predicting material type using a single-pixel multiband spectrometer [1]; and 3) predicting frictional force distributions by lateral pushing. These modes constitute the basis of our fully automatic system described in the following sections.

---

[1]SparkFun Triad Spectroscopy Sensor - AS7265x (Qwiic)

### 3.3 Entropy-guided interactions

In [2], qualitative experiments demonstrated that well-placed user clicks, especially in regions where the model is performing poorly, are the most beneficial in terms of improving segmentation quality. This observation was exploited in an automatic query generation framework, whereby an uncertainty-based sampling was used to actively propose pixel positions for the user to label. Similarly, we utilise softmax entropy to guide the physical interactions that the robot makes with the scene, encouraging interactions that are optimal in terms of information gain and thereby minimising the number of interactions required to produce optimal segmentations. Softmax entropy, $u_S$, is defined as [18]:

$$u_S = -\sum_{c=1}^{C} \hat{\mathbf{S}}_c[u, v] \log\left(\hat{\mathbf{S}}_c[u, v]\right), \tag{1}$$

with $\hat{\mathbf{S}}_c[u, v]$ the rendered semantic distribution and $C$ the number of categories.

### 3.4 Interaction feasibility

In order to integrate uncertainty-based sampling into fully-autonomous robotic behaviour, kinematic feasibility and the particular mode of interaction have to be considered. Queries deemed to be kinematically infeasible and/or unsuitable given the mode of interaction, are added to a query mask, preventing repeated testing. For example, top-down interactions require masking of candidate points not approximately perpendicular to the surface normal, while the converse is true for lateral interactions. Similarly, sensor type impacts interaction feasibility. For example, when performing spectroscopic measurements, we mask regions of high curvature to encourage measurements perpendicular to the target surface. More generally, we have found that interactions at object boundaries are generally undesirable, owing to a higher incidence of failures associated with misclassifications and object tipping. We have therefore investigated two approaches to reduce the probability of boundary interactions: 1) Gaussian filtering of the uncertainty maps and 2) uncertainty clipping. While Gaussian filtering yields reliable behaviour in scenes containing coherent objects, it performs poorly with overlapping objects (e.g. fabrics/clothing). In contrast, clipping the uncertainty map by setting values above a threshold to zero produces more reliable results. The robot focuses on regions of high uncertainty but ignores problematic regions of maximal uncertainty. There is a trade-off between the degree of boundary suppression and uncertainty map degradation. We determined empirically an optimal clipping threshold of 0.7 and kernel size of 41.

### 3.5 Semantic representation

In contrast to [2], the inputs to the semantic head of our neural-field MLP can be one of several physical properties, measured via apposite affordances and modes of interaction. Raw sensor measurements acquired by the robot need to be post-processed or converted into the target variable being predicted by the semantic head of the MLP. For binary prediction tasks (e.g. rigidity), this may be as simple as applying a threshold to the raw measurement. Multi-class target variables may require additional processing. For example, when predicting material type from a multidimensional spectrometer reading, we use a pretrained multiclass SVM classifier which outputs predefined material classes, which are then fed to the semantic head. In both scenarios, as in [19, 2] the semantic head of the MLP predicts a categorical value and can be optimised using cross-entropy loss.

We additionally demonstrate for the first time the prediction of continuous-valued target variables in the *semantic head* of the MLP, where the ground-truths are sparse, in contrast to the dense ground-truths used in the optimisation of the colour and density heads. For example, when predicting frictional force distributions, we feed the minimum (stiction) force required to move an object, directly to the semantic head and optimise using an $L1$ loss.

### 3.6 Single frame optimisation

During lateral pushing, interactions between the robot and the scene may introduce object displacements. This violates the static-scene assumption common to prior work in neural radiance fields. While this assumption allows [17, 2] to optimise over an expanding set of keyframes, a dynamic scene potentially invalidates historic keyframes, ultimately leading to errors in the reconstruction. We therefore clear the keyframe history and corresponding labels after each interaction in this mode. Our

experimentation has suggested that neural radiance representations possess some form of temporal memory characteristic over the labelled properties, whereby network weights adapt over time and maintain consistency with the dynamic scene, provided scene changes are comparatively small.

## 4 Experiments

A fundamental contribution of our work is the realisation of a fully-autonomous robot that operates in a real-world environment. We demonstrate the ability of our system to perform a series of autonomous experiments, using the aforementioned interactive modes, to discover and predict a variety of physical scene properties. We demonstrate the quantitative benefits of entropy-guided experimentation and, in the case of rigidity and material-type classification, compare segmentation performance against two state-of-the-art, class-agnostic segmentation techniques (see Sec. 4.2 for details). Finally, we refer the reader to our supplementary video for additional results.

### 4.1 Experimental setup

We use a Franka Emika Panda robot, anchored to a table on which a variety of objects are arranged (Figure 1). The robot is equipped with a Realsense D435 RGB-D sensor [20], tracked using the forward kinematics of the arm, which is controlled using ROS [21]. Prior to physical experiments, the robot builds a geometric reconstruction of the scene, to allow for collision-free motion planning. For this purpose, a set of RGB-D keyframes is captured over a series of random motions in order to optimise the 3D neural field, and subsequent querying of the network produces a collision mesh and normal map. All objects of interest are placed within reach of the robot arm and any points located beyond this range, or on the plane of the table, are automatically labelled 'table'.

### 4.2 Quantitative results

Rigidity and material-type prediction may be viewed as segmentation problems. While training any popular instance segmentation technique (e.g. Mask R-CNN [6]) on the object classes present in our scenes (e.g. material types), is likely to produce high-quality segmentations, one would need to repeat this training for each scenario. Therefore, instead of comparing against closed-set segmentation techniques, we consider two state-of-the-art class-agnostic instance segmentation approaches: 1) Mask R-CNN trained to perform class-agnostic segmentation [5] and 2) Unseen Clustering Network (UCN) [3] with RICE refinement [7] (see Sec. 2 for details). For each method, we perform instance segmentation on the keyframe and use the resulting instance mask to guide the robot-scene interaction. In particular, the robot takes a single sensor reading as near to the centre of each instance in the mask as is feasible and propagates the measurement to the rest of the region. The measurements are converted to categorical labels (binary rigidity or material-type) in the same manner as described in Sec. 3.5. We report the mean Intersection over Union (mIoU) averaged over the ground-truth labels.

Table 1 shows the quantitative performance comparison against the Mask R-CNN and UCN baselines for each scene. As expected, the baselines perform well for scenes 1 and 3, which contain geometrically-coherent objects and strong colour and depth cues. Scene 2, however, is considerably more challenging for the colour and/or depth-based baselines, characterised by a significant drop in performance. In contrast, our autonomous approach performs well for all three scenes, with comparable results to the baselines in Scenes 1 and 3 and significantly superior results in Scene 2.

### 4.3 Entropy exploration ablation study

Figure 2 illustrates material discovery in a complex scene containing wool (blue), cotton (yellow) and synthetic (green/pink) materials. Prior to physical measurements, there is high uncertainty (red) across the entire scene, while the final uncertainty map has high confidence (blue) throughout. We show the evolution of the uncertainty map through three consecutive interactions in Figure 3. The robot is guided to a high-entropy pixel (unfilled circle). On completion of the experiment, there is a clear, localised uncertainty reduction surrounding the target region (filled circle).

We observe that while the uncertainty in the localised region of measurement decreases, it often increases in more distant regions. As the model accumulates information, it continuously adapts its predictions and corresponding confidence, ultimately converging on an accurate representation. This

observation motivates the use of uncertainty as an exploration metric in neural implicit representations. To substantiate the benefits of entropy-driven exploration quantitatively, we conducted an ablation study comparing performance to random exploration. We compare the evolution of mIoU and false-confidence (where the model produces high-confidence but incorrect predictions) with an increasing number of interactions for each technique. Figure 5 demonstrates superior convergence rates for entropy-guided interaction in all three benchmark scenes in Figure 4 across both metrics.

## 4.4 Force measurement analysis

Figure 6 illustrates the stiction force maps produced by our framework for objects with uniform (top row) and non-uniform (bottom row) mass and friction distributions. Note that in each scene the objects are displaced following the pushes performed by the robot. The scene in the top row contains three cylindrical containers of varying mass and material. As desired, the resulting force renderings match the varying masses. This is potentially valuable information when planning for downstream manipulation tasks (e.g. distinguishing between full and empty containers). A key observation in Figure 6 is that the renderings for objects remain consistent despite displacement, demonstrating for the first time a memory quality in neural field representations.

In the bottom row of Figure 6, we demonstrate the ability of the robot to predict stiction force values reliably, even for complex geometries, with non-uniform mass and friction distributions. We substantiate this quantitatively in Figure 7, where the robot interacts with a non-uniform rectangular box containing a $5\,\mathrm{kg}$ weight at one end. We show that the output of our model, after three pushes, is comparable to that of a simple analytical physics model [22] with access to privileged information, including the contact surface area, mass distribution and friction coefficient.

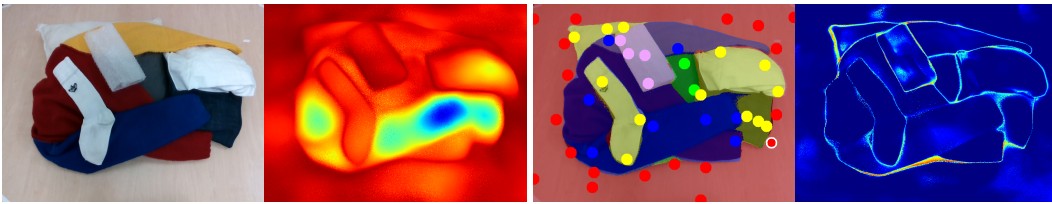

| Initial keyframe and uncertainty map | Final segmentation and uncertainty map |

Figure 2: Example material type segmentations using a spectrometer. 46 interactions were required to separate the pile of laundry into wool (blue), cotton (yellow) and synthetic (green/pink) materials. Red/blue signifies high/low uncertainty in the uncertainty map.

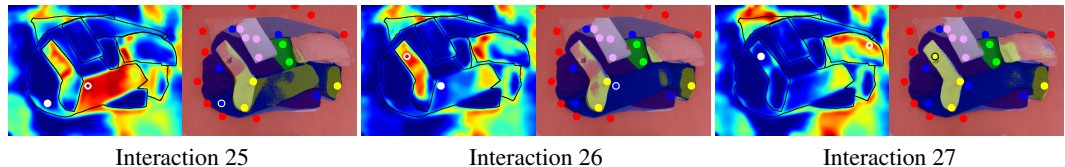

| Interaction 25 | Interaction 26 | Interaction 27 |

Figure 3: Demonstration of autonomous guidance over 3 consecutive interactions in Figure 2. From left to right, the interactions (unfilled markers) follow the highest uncertainty pixel. After interaction (filled marker), there is a localised reduction in uncertainty.

Table 1: Classification performance for different types of scene, (examples in Figure 4).

| Segmentation | Example | Ours | Mask R-CNN | UCN + RICE |
|---|---|---|---|---|
| Material | Scene 1 | $0.91 \pm 0.02$ | $0.92 \pm 0.02$ | $0.90 \pm 0.02$ |
| Material | Scene 2 | $0.89 \pm 0.03$ | $0.56 \pm 0.11$ | $0.56 \pm 0.10$ |
| Rigidity | Scene 3 | $0.91 \pm 0.04$ | $0.92 \pm 0.02$ | $0.91 \pm 0.02$ |

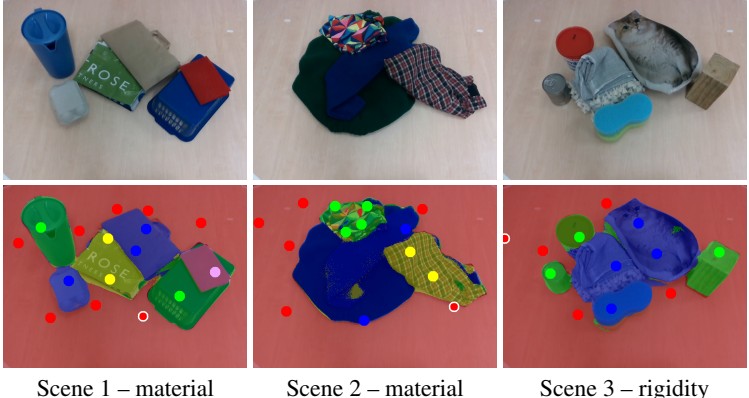

| Scene 1 – material | Scene 2 – material | Scene 3 – rigidity |

Figure 4: Our system can interact with and segment a variety of scenes.

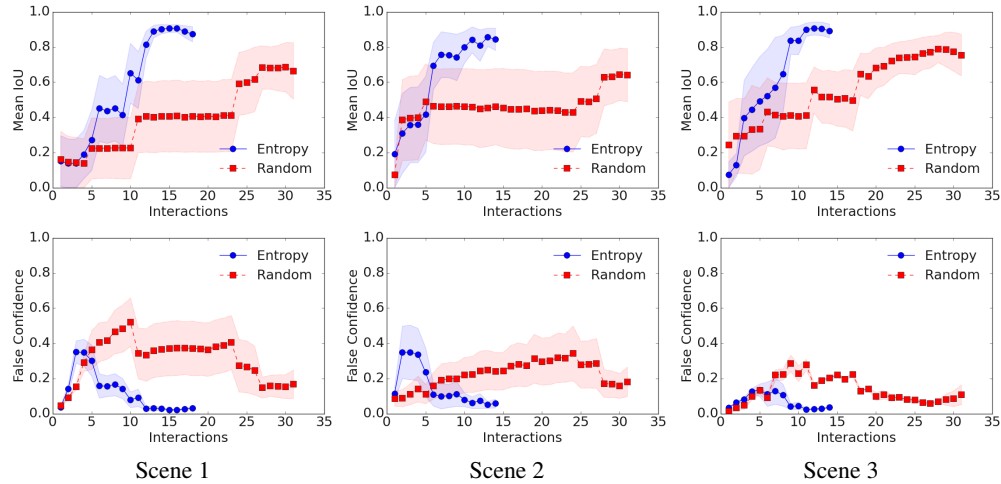

Figure 5: Comparisons of mean IoU (top) and false-confidence (bottom) vs. number of interactions for entropy-based and random exploration approaches.

## 5   Limitations

Several assumptions are made during the experimentation procedure: 1) rigid objects are never placed on top of flexible objects; 2) the SVM classifier has been trained on all materials and 3) pushed objects are both rigid and free to move across the tabletop.

The main limitations of our system relate to its self-awareness of errors. We only include examples in which all measurements and inferences are correct. In reality, this is not always the case, with each mode of measurement having an associated accuracy, and incorrect measurements impacting model performance. We observed that, although the model recovers after an erroneous measurement, it involves significantly more interactions. An alternate solution would be to associate a likelihood with each measurement and allow the model to adapt accordingly. Furthermore, with all scene interactions, there is a possibility of catastrophic disturbance, preventing the model from propagating labels effectively. This scenario was observed several times during experimentation, requiring a manual reset. Ideally, failures of this nature should be detected autonomously based on the corresponding surge in reconstruction loss.

Although several keyframes are captured during robot motion, interaction queries were only ever generated from the first keyframe, which in our setup was predefined as the initial viewpoint of the robot. This has the advantage of limiting the attention of the robot to what we deemed to be the most interesting part of the scene. However, it also meant the robot was unable to conduct

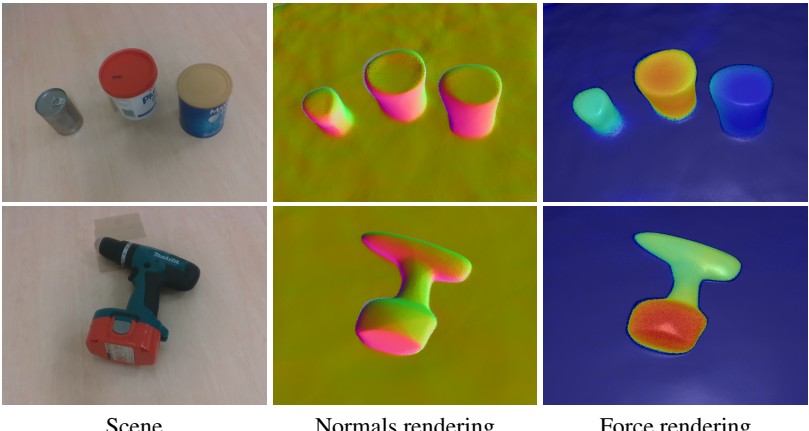

| Scene | Normals rendering | Force rendering |

Figure 6: Stiction force mapping. Top row: three cylindrical objects with uniform mass of (from left to right) $0.5\,\text{kg}$, $1.5\,\text{kg}$ and $0.1\,\text{kg}$. Guided by entropy, the robot applies a single push to each object measuring, stiction forces of $1.0\,\text{N}$, $3.0\,\text{N}$ and $0.2\,\text{N}$. Bottom row: power drill with non-uniform mass distribution. The final rendering was produced after a sequence of 3 pushes.

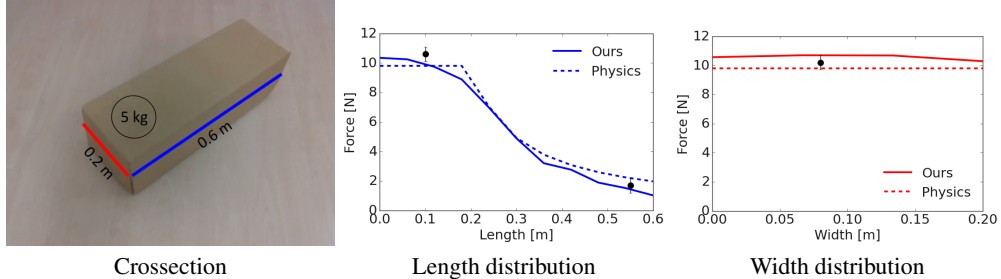

| Crossection | Length distribution | Width distribution |

Figure 7: Rendered stiction force distribution, compared to an analytical approach with privileged information, along the length (blue) and width (red) of a box with non-uniform density.

wider investigation; in particular, the back surfaces of objects or those regions occluded in the initial viewpoint. This was most evident in lateral pushing, where pushing from multiple angles could only be achieved with a careful choice of viewpoint and object positioning. A future improvement would address this by allowing queries across multiple keyframes, though the robot will still be constrained by its kinematic limits.

The sensitivity of force measurement is limited by the robot joint torque sensors and subsequent estimation of external torques and wrenches. This could be improved by using an external force-torque or tactile sensor. Using a spectrometer with greater bandwidth would allow for enhanced differentiation between material types. Finally, while we propose that our system facilitates the discovery of a large variety of physical scene properties, we only consider three specific scenarios in this paper.

## 6 Conclusions

By taking advantage of the unsupervised decomposition capabilities of a 3D neural field trained on a single scene, we have demonstrated a fully-automatic, real-world robotic platform that is able to build dense, accurate and physically meaningful representations of its environment. The fact that this can be achieved from scratch and in real-time, without any pre-training or external guidance, gives the method potentially wide applicability in robotic scene learning and understanding. In the future, we are excited to extend the system with new interactions, such as probing a scene with a gripper to build a grasping map, or using alternative sensors to measure properties such as temperature or surface roughness.

**Acknowledgments**

Research presented in this paper has been supported by Dyson Technology Ltd. We would like to thank Charles Collis at Dyson Technology Ltd. for his support and in enabling the secondment of Iain Haughton from Dyson Technology Ltd. to the Dyson Robotics Lab, Imperial College. We also wish to thank Barry Beard at Dyson Technology Ltd. for his help with using the spectroscopy sensor for material classification. Finally, we would like to thank Tristan Laidlow, Daniel Lenton, Kentaro Wada and Shuaifeng Zhi who engaged in research discussions as well as the reviewers who provided valued feedback on our work.

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
