# OpenReview forum: "Real-time Mapping of Physical Scene Properties with an Autonomous Robot Experimenter"
_robot-learning.org/CoRL/2022/Conference — CoRL 2022 Oral_

### Official Review · Reviewer_CQYj · 2022-07-31

**Originality:** Very Good
**Technical Quality:** Very Good
**Clarity Of Presentation:** Excellent
**Impact:** 4

**Recommendation:**

Strong Accept: I recommend accepting the paper and will argue for my recommendation even if other reviewers hold a different opinion.

**Summary:**

The paper proposes autonomous interactive experimentation with a scene in order to learn its properties, leveraging the ability of 3D neural field representations to generalise given as few as tens of examples. Experimental design is guided by entropy, maximising confidence under the constraints of the platform and environment. A workaround is included for the dynamics resulting from interactive experimentation. A generalisation of neural semantic labelling to continuous-valued semantics is included. The approach is demonstrated estimating rigidity, material, and frictional force distribution.

**Issues:**

The authors may wish to expand on the limitations section as per above comments, and consider releasing code and exemplar datasets. It would have been stronger to include more experimental results around force estimation, this could perhaps form the basis for an extension of the work.

**Quality Of The Limitations Section:**

Additional details required

**Reviewer Expertise:**

4: The reviewer is confident but not absolutely certain that the evaluation is correct

**Robotics Focus:**

Sufficient demonstration on hardware

**Strengths And Weaknesses:**

The paper makes important and timely advances in the joint representation of geometry and semantics using neural fields. Several contributions combine to yield an impressive demonstration: experimental design under constraints, dealing with dynamics, and encoding continuous semantics work together to enable a novel autonomous robotic explorer. This is a clever use of point sensors, allowing complex properties like hyperspectral signatures to be sensed without need for more expensive snapshot hyperspectral cameras, or more time-consuming pushbroom scanning.

The literature review covers the main related works well and relates the contributions of the present work. The paper is well represented and contains sufficient detail to allow replication by a skilled practitioner.

Visual results are included that qualitatively demonstrate and validate the approach. Quantitative results establish the efficacy of the experimental design with respect to naive approaches for segmentation and classfication accuracy. Force estimation is quantitatively compared to an analytic approach with privileged information, comparing favourably.

While some limitations are discussed, the paper misses discussion of some key limitations of the method. The underlying 3D representation discards the directional dependence captured by NeRF, for example, meaning the method should be expected to fail around visually challenging reflective and refractive objects. A robot experimenter can accumulate a vast and always-growing repertoire of experimental results... how to deal with this growing dataset is not discussed.

The paper would be strengthened through release of source code and datasets.

The force estimation results show only one example, a more comprehensive demonstration with multiple object types and aggregate statistics would be more convincing.

The references incorrectly render at least two acronyms in lowercase.

**Summary Of Recommendation:**

The paper is well presented and makes multiple important and timely advancements that will be of interest to a substantial part of the robotics community.

---

> ### Author Response · Authors · 2022-08-26
> **Response to Reviewer CQYj**
>
> We greatly appreciate the time you’ve put into reading our work and raising thoughtful comments. We have focussed on replying to the weaknesses and hope our following responses prove satisfactory.
>
> Thank you for your suggestion to expand our limitations section. Directional dependence and a growing dataset are limitations shared with the prior works of iMap and iLabel. Given the strict page limit, we would like to prioritise limitations linked directly to our work, however, if space allows, we will certainly consider adding these.
>
> Whilst we would like to release source code, unfortunately it is not currently possible due to IP implications.
>
> With regards to stiction force estimation, we would like to highlight the three examples shown in the paper. Figure 6 (top row) shows the interaction with 3 cylindrical objects with uniform mass distribution. The measured stiction force to mass ratios all result in friction coefficients measuring ~0.2, which is reasonable given the laminated surface on which the objects are pushed. In the same Figure 6, the bottom row shows the prediction of stiction force across a power drill, an object of non-uniform mass distribution. Finally, in Figure 7, the stiction distribution predicted for a box, again with non-uniform mass distribution, is compared favourably to an analytical model.
>
> Once again thank you for taking the time to read and critique our work, it was very pleasing to hear of your enjoyment in reading it.

---

### Official Review · Reviewer_yfuZ · 2022-07-31

**Originality:** Very Good
**Technical Quality:** Good
**Clarity Of Presentation:** Good
**Impact:** 4

**Recommendation:**

Weak Accept: I recommend accepting the paper, but will not argue for my recommendation if the majority of other reviewers have a different opinion.

**Summary:**

This paper proposes an approach for letting a robot learn about its surroundings through physical interaction. The approach builds upon an existing work for interactive 3D scene labeling ( Reference [2] in the manuscript). The method is largely agnostic to the segmentation task, as long as the property of interest can be efficiently sensed.

**Issues:**

I think the paper would benefit from a better discussion of previous work and the differences between the proposed approach and the iLabel work, upon which it builds. In addition, more ablation studies would improve the quality of the experimental section.

**Quality Of The Limitations Section:**

Limitations are addressed clearly

**Reviewer Expertise:**

3: The reviewer is fairly confident that the evaluation is correct

**Robotics Focus:**

Sufficient demonstration on hardware

**Strengths And Weaknesses:**

The main strength of this paper is its underlying idea: using proprioceptive experience to label physical properties of the world is much more scalable than having human annotators collect data. Some previous works (e.g. [1] and other references in the paper) proposed a similar idea, but the proposed setup differs in several aspects.
The proposed approach is also very interesting: connecting the two worlds of assisted labeling and learning by interaction is very interesting and well-suited to the problem. Of course, several technical changes have been required to make the approach work for the current problem. However,  I guess that several works in the future can benefit from this connection.
Finally, the work is quite well executed: results are provided on multiple segmentation tasks, showing the generality of the idea.

I see two main weaknesses in the manuscript. First, the information about the differences between the original Ilabel work and the proposed method is scattered and difficult to grasp. For example, the entropy-based formulation was already proposed by it, but it is not clearly mentioned. I would propose an overview paragraph explaining in detail the required changes that made the system work.
Related to this limitation, not enough ablation studies have been conducted. Ideally, each of the changes required by IMAP should be ablated with controlled experiments. This would give the reader a good grasp of what pipeline parts need improvement.

Overall, I think this is an interesting paper that could be of interest to the community. I also appreciate the extensive limitation section, which gives an honest overview of the current status of the idea of learning by interaction.

[1] Learning Depth With Very Sparse Supervision, Loquercio et al.

**Summary Of Recommendation:**

My recommendation is based on the interesting idea and approach. I could increase my score given more controlled experiments on the different aspects of the methodology.

---

> ### Author Response · Authors · 2022-08-26
> **Response to Reviewer yfuZ**
>
> We were very pleased to read your predominantly positive comments on our work and would like to convey our thanks and appreciation for your time in providing valuable feedback. We hope you find our following responses satisfactory.
>
> As suggested, we set about writing an overview paragraph explaining the required changes from iMap/iLabel to our work, however, after further consideration, we found this to be almost identical to lines 19 -> 42 in the introduction. As such, we would like to refer the reviewer to this section with the hope of providing clarification. In summary:
>
> We build upon iMAP and iLabel by exploiting an active, autonomous agent to remove the human from the loop entirely. We extend the predictive capabilities of the underlying MLP to include physical scene properties, which the robot autonomously queries from the scene. We make technical and implementational contributions in enabling our system to operate in a fully-autonomous manner on a physical agent to obtain rich, task-driven scene representations. Another significant change in our system is the method for camera tracking; given the robot embodiment, we are able to use kinematics as opposed to frame optimization. Synchronisation between image capture and joint values resulted in comparative construction of scene meshes to prior work, as can be seen in the renderings of Figures 1 and 6, with reduced computational cost.
>
> With regards to further ablation studies, thank you for your suggestion. Indeed further studies were considered, however many of these, including: depth / RGB weighting; choice of uncertainty metric; comparison to human labelling, were demonstrated in the prior works of iMap and iLabel.

---

### Official Review · Reviewer_beEy · 2022-07-31

**Originality:** Very Good
**Technical Quality:** Very Good
**Clarity Of Presentation:** Excellent
**Impact:** 4

**Recommendation:**

Strong Accept: I recommend accepting the paper and will argue for my recommendation even if other reviewers hold a different opinion.

**Summary:**

This paper shows a robot manipulator system equipped with a camera, force sensors at the joints, and a single-pixel multi-band spectrometer, that aims to learn physical properties of the table-top scene through active probing and interaction. The inferred properties include the material type of the object (from the spectrometer), the static friction force threshold required to move an object by lateral pushing, a rigidity prediction based on top-down poking. The paper builds on top of iMAP [1] and iLabel [2] for real-time neural scene representation and interactive labeling from few demonstrations.

[1] iMAP: Implicit Mapping and Positioning in Real-Time
[2] ILabel: Interactive Neural Scene Labelling


**Issues:**

I am hoping that the weaknesses I mentioned can be addressed easily during the revision period, as they are mostly related to presentation and writing, and do not require additional experiments.

**Quality Of The Limitations Section:**

Limitations are addressed clearly

**Reviewer Expertise:**

5: The reviewer is absolutely certain that the evaluation is correct and very familiar with the relevant literature

**Robotics Focus:**

Sufficient demonstration on hardware

**Strengths And Weaknesses:**

Strengths:

S1: In my opinion, the paper's main contribution comes from showing that fully autonomous interactive perception can be very effective in figuring out object properties with few interactions, for both rigid and deformable objects, such as clothes, even when neural scene representations are used. Active sensing is often tricky because random queries seem to catch up to active queries faster than hoped for, in some domains. It is informative to know that this is not the case here.

S2: The use of the spectrometer is something that rarely appears in robotics literature, but seems to be very helpful in categorizing material types. Although the material-type classifier was pretrained on all materials, I think it enables a range of possible applications in manipulation, and more roboticists should be aware of these possibilities.

S3: The paper is clearly written, with many descriptive figures, and an informative video.

S4: The limitation section is thoughtful, honestly written, and extensive. I applaud the authors for doing this well.

Weaknesses:

W1: The paper claims that its neural representation can handle dynamically changing scenes, for example "learning neural field representations in dynamic environments using dynamic-keyframe optimisation." This claim is slightly misleading, as the paper clarifies later on, as the history of old keyframes from iMAP gets dropped, and the neural representation is only valid if the changes in the scene are small, as is the case usually in the lateral pushing experiments. This is indeed clarified in section 3.6, but there is an incongruence between the claimed contributions in the introduction and the strict conditions under which they hold. This is a hard problem, understandably, but I would have appreciated more clarity about assumptions in the claimed contributions.

W2: Interaction queries are generated only from the first viewpoint, and are not updated as the robot acquired more measurements. This slightly undermines the goal of the paper to perform fully active perception, since dynamic scene changes are not accounted for, but it is not a deal-breaker. It just means that the active query selection is not performed optimally.

W3: Entropy-based active perception is claimed as a contribution, but entropy has been used as an acquisition function for many learning systems and labeling tasks, including in robotics. While this is a key component of the paper, and seems to work well, I don't think it has to be claimed as a novel contribution. The fact that it works well is informative enough for the reader.

W4: Clarification needed, what do the error bars in Fig 5 represent? And why are the error bars for the random selection strategy not decreasing much over time?

W5: Nitpick, but, it would be better if the 3d plots in Fig 1 were rotated to match the orientation of the scene.

W6: It would be useful to mention the exact model of the spectrometer used in the paper.

**Summary Of Recommendation:**

I find this paper very exciting, and I am recommending it be accepted because of strengths S1 and S2. I also very much enjoyed the fact that the paper was not limited to rigid objects.

---

> ### Author Response · Authors · 2022-08-26
> **Response to Reviewer beEy**
>
> Thank you for taking the time to read and critique our work, it was very gratifying to hear of your enjoyment in reading it. We have made several changes to the paper (highlighted in yellow) in the hope of addressing some of your weaknesses, and would like to provide the following responses.
>
> W2: A static scene is assumed for both the top-down poking and spectroscopy experiments, in which keyrames, gathered over time, are used to train the network. You are correct in suggesting that interaction points are only generated in a region covered by the first view point. However, we would argue their selection is optimal, since queries are generated from rendering the most up-to-date neural representation.
>
> W3: We agree that entropy-based active perception is not a contribution and have removed these claims. However, we maintain the novelty in achieving real-world, autonomous behaviour using a single network to drive entropy-guided interaction, collision avoidance, kinematic feasibility checking and path planning.
>
> W4: The mean IOU is calculated as an average across the semantic classes, the error bars represent the variance on this value. As the mean IOU approaches 1, due to the upper limit, the variance will decrease. The random method doesn’t perform well enough to substantially reduce the variance.
>
> W5: Thank-you for this suggestion and we have indeed considered this. However, since large portions of the external viewpoint lie outside the ROI of the robot (eg. table extremities / walls) the network rendering is poor and the resulting image messy by comparison.

---

### Official Review · Reviewer_jBNQ · 2022-08-01

**Originality:** Very Good
**Technical Quality:** Very Good
**Clarity Of Presentation:** Very Good
**Impact:** 4

**Recommendation:**

Weak Accept: I recommend accepting the paper, but will not argue for my recommendation if the majority of other reviewers have a different opinion.

**Summary:**

This paper proposes to leverage autonomous robots to explore and collect self-supervised data for training neural fields. Specifically, the authors show that neural fields can be self-supervised to predict materials and force distribution in a real-world setting.

**Issues:**

I wish the authors address my concerns about **Clarity** within the Strengths And Weaknesses section.


**Quality Of The Limitations Section:**

Limitations are addressed clearly

**Reviewer Expertise:**

4: The reviewer is confident but not absolutely certain that the evaluation is correct

**Robotics Focus:**

Sufficient demonstration on hardware

**Strengths And Weaknesses:**

### Strengths

**Originality**

I think the paper has high originality. The prediction of force distribution is novel and interesting.

**Quality**

The proposed method is technically sound and applied appropriately. The experiments are thorough and the video results look promising.

### Weaknesses

**Clarity**

Although I think the paper is novel, several contributions listed in the Introduction are overclaimed, and not enough related works are cited:
- Contribution 1 “the first fully autonomous, real-time neural scene labeling robot capable of operating in the real-world” This claim is simply not true. Both [2] and [3] and earlier works they cited have applied autonomous, interactive perception to various problems such as articulated parts discovery. I recommend authors read [4] and refine the claim.
- Contribution 3 “dynamic-keyframe optimization” basically means discarding all the frames after a mode of interaction.
- Contribution 4 “prediction of continuous-valued semantics using a neural-field model” I am not sure why this is a contribution. The most popular values predicted by neural fields such as RGBs and signed distances are all continuous.

**Citation**

For the paragraph starting at line 78, I suggest authors also cite NICE-SLAM [1].

**Typo**

- Line 57: interactive perception, affordance learning and online scene understanding. -> interactive perception, affordance learning, and online scene understanding.
- Line 136: The robot to focuses on -> The robot focuses on
[1] NICE-SLAM: Neural Implicit Scalable Encoding for SLAM, Zhu et al.
[2] Structure From Action Learning Interactions for Articulated Object 3D Structure Discovery, Nie et al.
[3] Ditto Building Digital Twins of Articulated Objects from Interaction, Jiang et al.
[4] Interactive Perception: Leveraging Action in Perception and Perception in Action, Bohg et al.


**Summary Of Recommendation:**

I recommend Weak Accept but wish the authors to consider my opinion in the Weaknesses section and rephrase the contributions.

---

> ### Author Response · Authors · 2022-08-26
> **Response to Reviewer jBNQ**
>
> It’s very pleasing to hear that you find our work both novel and interesting and we appreciate the time you put into providing relevant feedback. From this we have made several changes to the paper (highlighted in yellow) and hope you find our following responses satisfactory.
>
> As suggested, we have endeavoured to improve the clarity of our contributions and hope the highlighted changes in the attached paper are sufficient.
>
> In particular, with regards to claim 1, we have emphasised that our work is the first real-time neural scene labelling robot trained from scratch in real time with no human intervention. In contrast, [2] uses a voxelised representation with no NeRF component and [3] is pre-trained offline and requires manual intervention for defining interactions both in simulation and in the real world (see lines 48).
> Regarding claim 3, we agree that the term dynamic keyframe optimisation is misleading. We have reworded these claims accordingly, emphasising what we believe to be an observational contribution in the recognition of the apparent temporal memory characteristics of neural field MLPs - which have not been noted before (see lines 51-52).
>
> Claim 4 has been clarified to stress what we believe to be the first work predicting continuous valued semantics from sparse ground truths, in contrast to the prediction of colour and occupancy, where the reference ground truth used in optimisation is by definition dense (see lines 53-54, 156-157).
>
> Finally, with regards to the addition of a citation in line 78, this paragraph discusses prior work in interactive online scene labelling, of which iLabel is the prior state of the art. We argue that [1] is purely a SLAM system with no labelling component and therefore is not relevant here. iMAP is cited here because it forms the basis of both iLabel and our system.
>
> Thank you for pointing out typos, these have all now been fixed.

---

### Meta-Review · Area_Chair_Ag4n · 2022-08-13

**Recommendation:** Accept (Poster)
**Confidence:** 4

**Metareview:**

This paper proposes an approach leveraging 3D neural implicit scene representations to autonomously infer rigidity, material, and frictional force distributions in real settings.

Overall, the reviews are positive. Reviewers agree that the presented ideas are compelling, but point to several areas (e.g. assumptions, limitations, ablations) that need additional clarification – particularly with respect to overclaiming and consolidating contributions. Authors are encouraged to respond to reviewers' comments and questions.

Update:

All reviewers agree that the paper is well-presented and should be accepted; the ideas are timely, and will be interesting to the robotics community. The authors did not submit a rebuttal – several reviewers re-raised concerns that were not addressed during the rebuttal phase, including a discussion on key limitations (visually challenging reflective and refractive objects: a key advantage of NeRF vs classic RGB-D fusion), as well as providing more clarity on assumptions being made for claimed contributions with respect to prior work. Authors are strongly encouraged to address the concerns raised in the original reviews for the camera-ready version.